# On the estimation of the fill rate for the continuous (*s, S*) inventory system for the lost sales context

**Ester Guijarro, Eugenia Babiloni** *, **Manuel Cardós**

Universitat Politècnica de València, Valencia, Spain

* mabagri@doe.upv.es

**Data Availability Statement:** All relevant data are within the manuscript and its Supporting Information files.

**Funding:** Initials of authors that receive the grant: E.G. and E. B. Grant number: SBPLY/19/180501/

## Abstract

In the continuous review reorder point, base-stock (*s, S*) policy, the replenishment order is launched when the inventory position reaches the reorder point, *s*. It is commonly assumed that the inventory position is exactly equal to the reorder point at the moment the order is launched, when actually it could be lower at that moment. This implies neglecting the possible undershoots at the reorder point, which has a direct impact on the calculation of the expected shortages per replenishment cycle. This article presents a method for an exact calculation of the fill rate (fraction of demand that is immediately satisfied from shelf) which takes explicit account of the existence of undershoots and is applicable to any discrete demand distribution function in a context of lost sales. This method is based on the determination of the stock probability vector at the moment the replenishment order is launched. Furthermore, neglecting the undershoots is shown to lead to an overestimation of the fill rate, particularly when we move farther away from the unitary demand assumption. From a practical point of view, this behaviour involves underestimating the base-stock level, *S*, when a target fill rate is set for its determination. The method proposed in this paper overcomes these shortcomings.

## Introduction

Inventory management systems are designed according to a measure of system performance, based either on costs or on the level of customer service. Since the costs associated with the system are difficult to estimate accurately [1,2] the service model is the most commonly used in practice. In this management context, one of the most widely used service measures is the fill rate (β further on) which is traditionally known as the fraction of total demand that is delivered from available stock without shortages [3]. Recently [4] identify several fill rate expressions, including the traditional one. This interesting metric not only reveals stockout situations, but also gives information on the size of the unmet demand [5,6] as it is calculated as the ratio between the satisfied demand and the total demand during a replenishment cycle. This paper focuses on the exact estimation of the fill rate when the system is continuously reviewed for the lost sales case that implies that unfilled demand is lost.

000151. Funder: European Regional Development Fund and Junta de Comunidades de Castilla-La Mancha (JCCM/FEDER, UE). URL: https://fondosestructurales.castillalamancha.es/ The funders had no role in study design, data collection and analysis, decision to publish, or preparation of the manuscript.

**Competing interests:** The authors have declared that no competing interests exist.

Continuous review inventory policies are very common in practice. The main characteristic of this type of policy is that the status of the inventory is known at all times, so that any change in the stock level is reported immediately. Basically, once the inventory position (i.e. on-hand stock + on-order stock—backorders) drops to the reorder point, *s*, or lower, a new replenishment order is launched and received *L* periods later. Such a replenishment order may be constant and equal to *Q*, as is the case of a system (*s*, *Q*), or variable until the base-stock level, *S*, is reached, and thereafter managed by means of a system (*s*, *S*). This research is dedicated to the (*s*, *S*) policy given that this policy is normally used in practice to manage strategic items [7]. This policy entails an additional difficulty: to exactly reach the reorder point, the demand must be necessarily unitary. If this is not the case the inventory position may not be exactly at the reorder point, but a certain amount below it, called deficit or undershoot, which is difficult to quantify. Therefore common derivations of a (*s*, *S*) policy are based upon the assumption that the undershoots at the reorder point are negligible [6]. However, neglecting undershoots may lead to higher inventory costs or a lower service level than desired [8,9]. This is particularly true when managing intermittent, lumpy or slow-moving items for which the unitary demand assumption is not appropriate [10] although undershoots can also occur with non-intermittent demand as long as the unit sized demand assumption is not met.

Authors who take explicit account of the undershoots at the reorder point aim to estimate the undershoot distribution function in such a way that the expected value of the undershoots is incorporated into the service measures. In the literature we find several approaches to the calculation of the undershoots based on asymptotic results or the renewal theory (see, for example [5,11–14], among others) or computing probability functions and the expected value of the undershoots [9,15,16]. Our approach, on contrary, instead of deriving an approach of the distribution function of the undershoots, considers that the inventory position when the reorder point is reached may take any feasible value (i.e. from 0 to *s*). This entails not neglecting the undershoots at the reorder point and implies the difficulty of accurately calculating expected values of the on-hand stock at order delivery and when the reorder point is reached (exactly or exceeded). This approach makes our research innovative.

Regarding the fill rate estimations in service models, there are only a few authors that explicitly consider the presence of undershoots. The studies by [5,11,17–19] suggest methods to compute the fill rate, however they are only for the backordering context. Under a lost sales context only [15], present an approximate method for the fill rate in a continuous review policy (*s*, *S*) that includes the expected undershoot value that is obtained from an approximate distribution function.

Summarizing, on the one hand, the (*s*, *S*) policy is widely used in practice but assumes that undershoots at the reorder point are negligible. This fact may introduce deviations on service measures such as the fill rate. On the other hand, there is no optimal approach to estimate the fill rate for the lost sales context, which has been hardly discussed in the literature [20]. Despite the fact that lost sales case is frequent in many sectors most of the studies are based on the total backordering assumption and more research is needed assuming the lost sales case [21]. The objective of this paper is to bridge that gap by proposing an exact method to estimate the fill rate. To address this objective, we will analyse the different issues to be taken into account for the calculation of the fill rate in the (*s*, *S*) policy and illustrate the bias that neglecting undershoots introduces in the model.

The remainder of this paper is organized as follows. Section 2 describes the inventory system, basic notation, assumptions and the description of the problem. Section 3 is dedicated to explain the mathematical derivation of the fill rate when undershoots are neglected and the exact method to compute the fill rate which is the main contribution of this paper. Section 4 presents the experimental design, discussion and practical implications of using a desired fill

rate to determine the base-stock level. Finally, the main conclusions of this paper are considered in Section 5.

## Problem formulation and description

### System description, notation and assumptions

We consider a single echelon, single item inventory system where demand is stochastic, stationary and i.i.d., and modelled by any discrete distribution function. Note that assuming discrete demand is closer to real world companies where continuous demand is rarely frequent, especially for intermittent items, such as spare parts [22].

Additionally, we assume that:

(i) The replenishment order is received after a constant and known lead time, *L*, and is added to the inventory at the end of the period in which it is received;

(ii) Demand is fulfilled with the on-hand stock at shelf.

(iii) Unfulfilled demand is lost.

(iv) Only one outstanding order is launched within any given period, which implies that $s < S - s$. Note that if it is not the case, the numerical difficulties are insurmountable [5], especially for the lost sales case [15,20].

Fig 1 shows the evolution of stock levels in a (*s*, *S*) inventory policy when the system is not out of stock (a) and when it is out of stock (b) for the lost sales case. Notation in it and in the rest of the paper is as follows:

*s* = reorder point, ROP (units),

*S* = base-stock level (units),

*L* = lead time for the replenishment order (time),

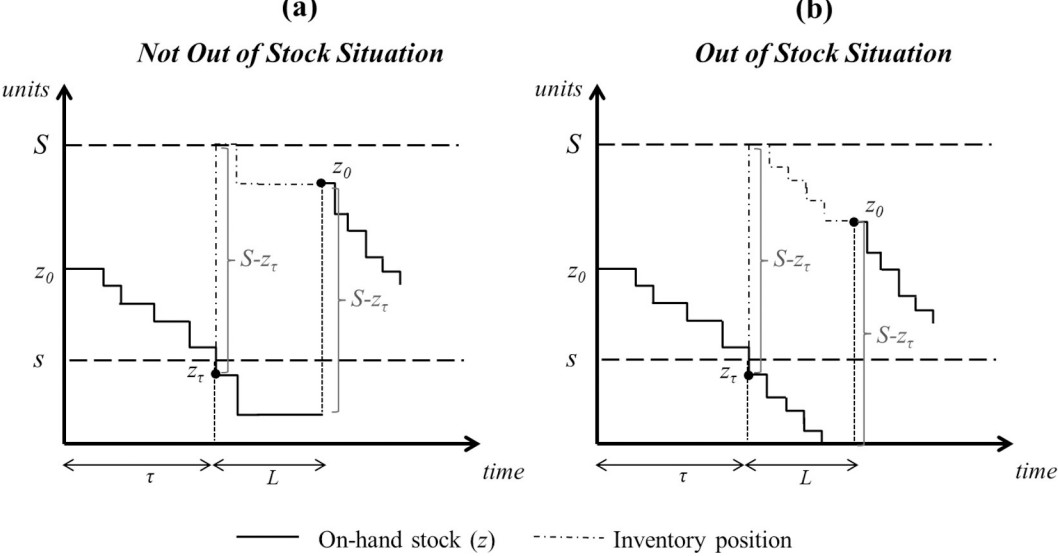

**Fig 1.** Evolution of the stock in a (*s*, *S*) inventory policy and lost sales when the system is not out of stock (a) and when it is out of stock (b).

$\tau$ = number of periods from the beginning of the cycle to the ROP (time),

$z_0$ = on-hand stock at the beginning of the cycle and at order delivery (units),

$z_\tau$ = on-hand stock when the reorder point is reached and the replenishment order is launched (units),

$d_t$ = demand at instant $t$ (units),

$D_\tau$ = accumulated demand from the beginning of the cycle to the ROP (units),

$D_L$ = accumulated demand during the lead time (units),

$f_t(\cdot)$ = probability mass function of demand at $t$,

$F_t(\cdot)$ = cumulative distribution function of demand during $t$ periods,

$h_t(\cdot)$ = probability mass function of $D_\tau$,

$H_t(\cdot)$ = cumulative distribution function corresponding to $g_t(\cdot)$,

$X^+$ = maximum $\{X, 0\}$ for any expression $X$.

## Problem description

By definition, the (*s*, *S*) policy implicitly states that both the replenishment cycle, i.e. time elapsed between two consecutive order deliveries, and the order quantity are variable. These two characteristics mean that mathematical approaches to characterize the policy are truly difficult to implement in practice. For example [23], points out that *"the values of the control parameters are set in a rather arbitrary fashion"* and the most common approach consists of assuming that all demand transactions are unit sized or, with the same consequence, that undershoots at ROP are not possible. It supposes that the inventory position always exactly reaches the ROP and therefore the order quantity is constant and equal to *S-s*, being equivalent to the policy (*s*, *Q*) with *Q* = *S-s* [24]. However, in practical environments it is uncommon to find items whose demand is unitary, which makes it necessary to consider the presence of undershoots when determining optimal parameters of the inventory system.

Another implication relates to the calculation of the total demand during the replenishment cycle. One of the main features of the continuous review policies is that there is no knowing when the reorder point will be reached. As a result, the replenishment cycles are variable, which, in a stochastic demand context, hinders the calculation of the demand distribution from the order delivery (beginning of cycle) until the reorder point is reached. Nevertheless, if the undershoots are neglected, the demand consumed from the beginning of the cycle until the reorder point is reached can be easily calculated as the difference between the on-hand stock at the beginning of the cycle and the ROP.

However, what is the cost of neglecting undershoots? The answer is related to the expected value of stockouts (also known as expected shortage) during the replenishment cycle. The assumption that the replenishment order is launched at the precise moment the reorder point is reached implies that the on-hand stock at launch is greater than it actually is, since $z_\tau < s$, in such a way that there is a higher probability of stockouts than one might expect. In the following sections we describe the bias that neglecting undershoots introduces on the fill rate as a consequence of overestimating the stockout probability of the system and also propose an exact method to estimate it that considers the existence of undershoots at ROP.

## On the fill rate estimation in *(s, S)* systems and lost sales

The fill rate is the fraction of demand that is immediately fulfilled from on-hand stock (at shelf). The most common approach to estimate this service measure consists of computing the complement of the ratio between the expected shortage per replenishment cycle (*ESPRC*) and the total expected demand per replenishment cycle (*EDPRC*), i.e.:

$$\beta = 1 - ESPRC/EDPRC \tag{1}$$

where *EDPRC* can be split into the accumulated demand from the beginning of the cycle until the ROP is reached, $D_\tau$, and the accumulated demand from the moment an order is placed until its delivery, $D_L$.

## Fill rate estimation neglecting the undershoot at ROP

This approach is based on neglecting the undershoot at ROP which leads to the assumption that the system always exactly reaches the reorder point. This assumption directly implies that demand is always unit sized and leads to the consideration of two important issues: (1) we always order the same amount. Note that the replenishment order is placed when the system reaches the ROP and therefore, equal to *S-s*. Indeed, the system becomes a (*Q*, *s*) system with *Q* = *S-s*; and (2) the system will only be out of stock during the lead time. Fig 2 shows the evolution of stock levels when undershoots are neglected. Both features (1) and (2) can be clearly appreciated.

When considering the effect of neglecting undershoots on the fill rate estimation, we see that this greatly reduces the mathematical complexity of it, since knowing the probability distribution of stock levels at the precise moment when the ROP is reached is probably the most complicated issue of its calculation. If the stock exactly reaches the ROP and the system is only out of stock if $D_L > s$, then the expected shortage per replenishment cycle is straightforwardly:

$$ESPRC = \sum_{i=s+1}^{\infty} (i - s) \cdot f_L(i) \tag{2}$$

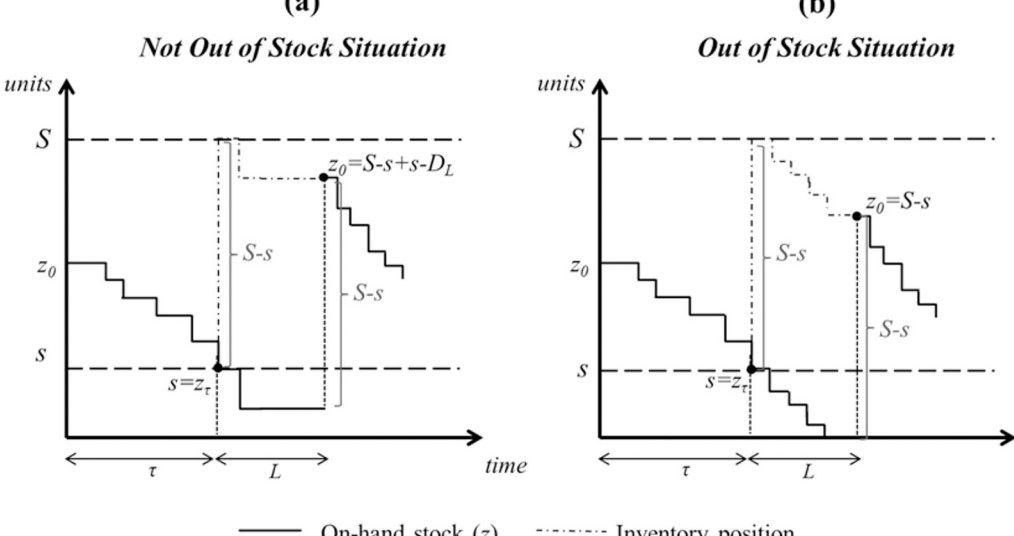

**Fig 2. Evolution of the stock in a (*s*, *S*) inventory policy when undershoots are neglected.**

To compute the expected total demand per replenishment cycle (*EDPRC*) we add up the accumulated demand from the beginning of the cycle until the ROP is reached, $D_\tau$, and the accumulated demand over the lead time, $D_L$. The latter is easily computed when the demand distribution is known. However, to compute $D_\tau$, we need to know the on-hand stock balance at the beginning of the cycle, i.e. at order delivery, which, in a lost sales context, is obtained thus $z_0 = S - s + E[s - D_L]^+$. Therefore, taking into account that $D_\tau$ has to guarantee that the *ROP* is exactly reached, then $D_\tau = z_0 - s = S - 2s + E[s - D_L]^+$.

Consequently, the expression to estimate the fill rate when undershoots are neglected, named as "classic" further on, for the lost sales case that applies to any discrete distribution is:

$$\beta_{Classic} = 1 - \frac{\sum\limits_{i=s+1}^{\infty} (i - s) \cdot f_L(i)}{S - 2s + \sum\limits_{j=0}^{s} (s - j) \cdot f_L(j) + \sum\limits_{k=0}^{\infty} k \cdot f_L(k)} \tag{3}$$

## General fill rate estimation considering the undershoot at ROP

To estimate the fill rate without any assumptions regarding undershoots at ROP (or demand size) means we have to address three important issues. Firstly, replenishment orders are variable and depend on the stock level when the ROP is reached, i.e. $Q = S - z_\tau$. In a stochastic context, to compute the expected value of $z_\tau$ we need to estimate its probability vector using transition matrices over the cycle. Secondly, the estimation of the total demand and concretely $D_\tau$ is not straightforward and also depends on the expected values of stock levels and on the unknown number of periods until the ROP is reached, $\tau$. Thirdly, stockouts may occur not only during the lead time, but also beforehand, in such a way that it occurs at the very moment the ROP is reached. In this section we address how these three issues affect the estimation of the fill rate and how to solve them to finally obtain an exact expression to compute it when undershoots are considered for the lost sales case and discrete i.i.d. demands.

The importance of knowing the stock levels over the replenishment cycle to compute the fill rate is based on the fact that there is no way of knowing the satisfied demand or, therefore, the expected backorders, without knowing the probability distribution of the on-hand stock levels when the ROP is reached. The stock level of policy (*s*, *S*) satisfies the Markov property, i.e. the future state only depends on the current situation. As a result, on-hand stock levels can be represented by a Markov process and the probability transition matrix of the on-hand stock between two consecutive replenishment cycles $\overline{\overline{M}}$ needs to be determined so that:

$$\lim_{n \to \infty} \overline{P(z)} \cdot \overline{\overline{M}}^n = \overline{P(z_0)} \tag{4}$$

where $\overline{P(z)}$ is an arbitrary vector and $\overline{P(z_0)}$ the principal left eigenvector of $\overline{\overline{M}}$, i.e. the probability vector of the on-hand stock levels at order delivery.

In order to calculate $\overline{\overline{M}}$ it is necessary to know the probability transition matrices from the beginning of the cycle until the ROP is reached, $\overline{\overline{M_\tau}}$, and from the moment the ROP is reached and the order is launched until it is received *L* units of time later, $\overline{\overline{M_L}}$, in such a way that $\overline{\overline{M}} = \overline{\overline{M_\tau}} \cdot \overline{\overline{M_L}}$. Once $\overline{\overline{M}}$ is calculated, we can obtain $\overline{P(z_0)}$ resulting from the convergence of the expression (4). The probability vector of the on-hand stock at ROP is calculated using the following expression: $\overline{P(z_\tau)} = \overline{P(z)} \cdot \overline{\overline{M_\tau}}$. How to obtain $\overline{\overline{M_\tau}}$ and $\overline{\overline{M_L}}$ is detailed in S1 Appendix.

The second important issue is related to the calculation of the expected demand during the cycle, *EDPRC*, since it is mathematically complicated to know the accumulated demand until

the inventory position exceeds the ROP, $D_\tau$. As mentioned above, the main problem resides in the fact that the number of units of time up to this moment, $\tau$, is unknown. To address this problem, we correct the derivation presented by [10] for the (*s*, *Q*) policy and adjust it to cover the special features of the (*s*, *S*) policy. S2 Appendix is dedicated to the mathematical derivation of the demand distribution from the beginning of the cycle until the ROP: $h(\cdot)$. Once $D_\tau$ is estimated, *EDPRC* is the sum of $D_\tau$ and $D_L$.

Thirdly, to compute the expected shortage per replenishment cycle, *ESPRC*, it is necessary to consider that stockouts may appear at the same instant the ROP is reached. Thus, two stockout situations are identified: (1) the on-hand stock is zero and the net stock is negative at the very moment the ROP is reached (Fig 3A); and (2) the on-hand stock is depleted once the ROP is reached (Fig 3B). In the case of (1):

$$ESPRC(1) = \sum_{z_0=S-s}^{S} \left( P(z_0) \cdot \sum_{\vartheta=z_0+1}^{\infty} (\vartheta - z_0) \cdot h(\vartheta|z_0) \right) \tag{5}$$

While in the case of (2):

$$ESPRC(2) = \sum_{z_\tau=0}^{s} \left( P(z_\tau) \cdot \sum_{\xi=z_\tau+1}^{\infty} (\xi - z_\tau) \cdot f_L(\xi) \right) \tag{6}$$

Finally, the exact fill rate in a context of discrete demand and lost sales considering undershoots is obtained through the following expression:

$$\beta_{Exact} = 1 - \frac{\sum_{z_0=S-s}^{S} \left( P(z_0) \cdot \sum_{\vartheta=z_0+1}^{\infty} (\vartheta - z_0) \cdot h(\vartheta|z_0) \right) + \sum_{z_\tau=0}^{s} \left( P(z_\tau) \cdot \sum_{\xi=z_\tau+1}^{\infty} (\xi - z_\tau) \cdot f_L(\xi) \right)}{\sum_{z_0=S-s}^{S} \left( P(z_0) \cdot \sum_{\vartheta=1}^{\infty} \vartheta \cdot h(\vartheta|z_0) \right) + \sum_{\xi=1}^{\infty} \xi \cdot f_L(\xi)} \tag{7}$$

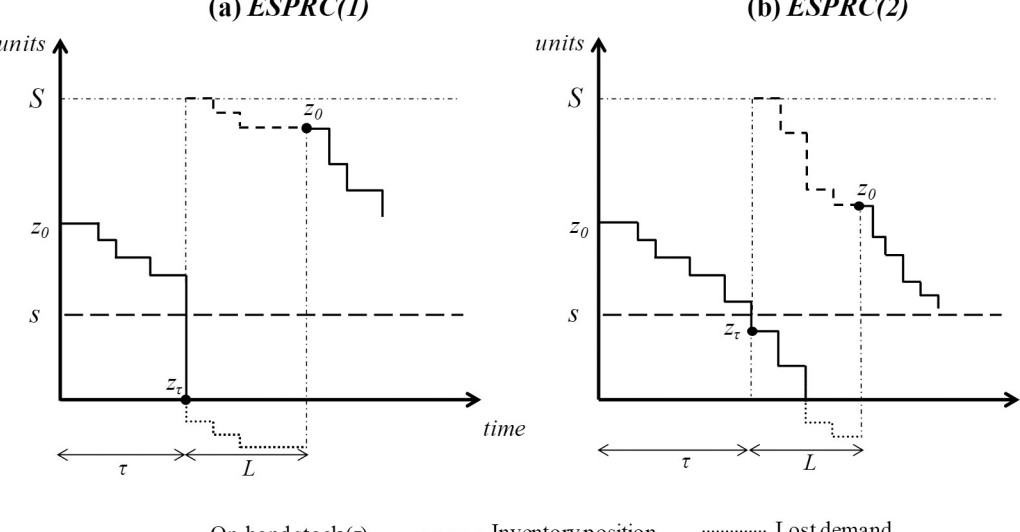

**Fig 3. Evolution of a (*s*, *S*) inventory policy that illustrates the *ESPRC(1)* and *ESPRC*(2) cases.**

## Experimental design, discussion and practical implications

### Experimental design

This section illustrates the performance of $\beta_{Classic}$ and $\beta_{Exact}$ against the simulated fill rate, $\beta_{Sim}$, which is computed as the complement of the ratio between the expected shortage and the total demand per replenishment cycle when considering 20,000 consecutive periods. This simulation uses the data from Table 1 for the cases whenever $\beta_{Sim} > 0.50$ with Pure Poisson demands which fulfil the smooth and intermittent categories according to [25] categorization framework. Thirty replications were applied to each case. The averages of the values obtained are the ones to be used as the final $\beta_{Sim}$.

### Results and discussion

Fig 4A and 4B show the comparison between $\beta_{Classic}$ and $\beta_{Sim}$ and between $\beta_{Exact}$ and $\beta_{Sim}$ respectively (S1 Table). As can be seen, $\beta_{Classic}$ tends to overestimate $\beta_{Sim}$. This behaviour was to be expected if we analyse the impact of neglecting the undershoots. For example, if the inventory position and the on-hand stock are one unit above the ROP at a given moment *t*, and an order for 2 units is received at *t+1*, a replenishment order is launched when the inventory position and the on-hand stock are equal to exactly *s-1* units. Therefore, the available on-hand stock remaining on the shelves to meet demand during the lead time is one unit less than that estimated when the undershoots are neglected. Thus, $\beta_{Classic}$ is higher than the actual value and therefore the stockouts are larger than expected.

This example goes to show why neglecting the undershoots introduces a significant bias into the classic fill rate calculation, a bias which will grow as the demand increases (further removed from the unit). Table 2 details the average and standard deviation of the errors made by $\beta_{Classic}$ as compared to $\beta_{Sim}$ with regard to the value of λ. What can be seen in Table 2 is that when the demand rate increases, meaning it is more difficult to assume the unitary demand hypothesis, the average and deviation of the errors made by $\beta_{Classic}$ increase. However, this example illustrates that undershoots appears also for non-intermittent demand, which justifies that undershoots should be taken into account explicitly in the calculation of the fill rate.

### Practical implications

Service measures are usually used to determine the control parameters of the inventory policy. In practice, for one specific SKU the base-stock is determined to fulfil a target fill rate taking into account its criticality. At this point, if the method used to calculate the fill rate is one which systematically overestimates its real value, as occurs in the case of $\beta_{Classic}$, the base-stock is lower than that required to reach the target fill rate, and therefore the system will be less protected against stockouts than managers might expect. Table 3 presents an illustrative example that helps to demonstrate the idea. In this case, if a target fill rate of 0.9 is set, according to the classic estimation of the fill rate, a base-stock of 7 units is deemed enough to reach the target fill rate, when actually it is 10 units that are needed to guarantee it. Another interesting detail that is shown in Table 3 is related to the fact that small differences in the fill rate can lead to

**Table 1. Set of data.**

| |
| --- |
| Lead Time *L* = 2, 3, 4 |
| Base-stock *S* = 5, 6, 7, 8, 9 |
| Reorder point *s* = 2, 3, 4 |
| Demand Variability (Poisson distributed) λ = 0.1, 0.5, 0.75, 1, 1.25, 1.5 |

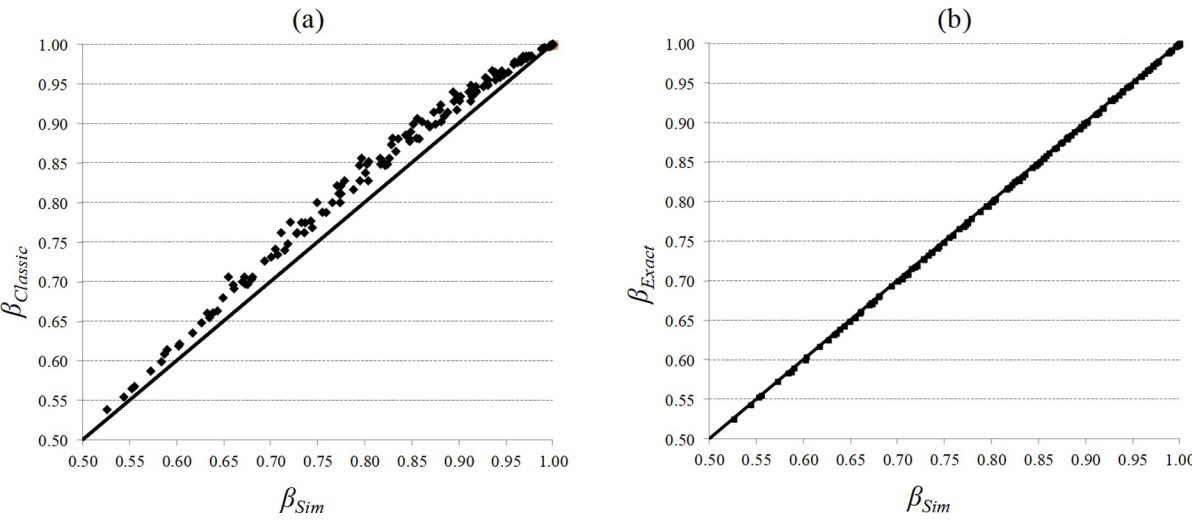

Fig 4. $\beta_{Classic}$ and $\beta_{Exact}$ vs. $\beta_{Sim}$.

important differences in the value of S. For the same example, a difference between both fill rates equal to only 0.01 ($\beta_{Classic}$ = 0.90 and $\beta_{Classic}$ = 0.91) leads to a difference of 3 units in the base-stock level. Therefore, despite the difference between the values of the fill rate seeming to be insignificant in some cases, it may become extremely important in the context of the base-stocks values and managers have to be forewarned about this fact.

One of the traditional problems in inventory management consists of how to compute accurately the optimal value of s and S simultaneously. This problem may be solved using a cost approach trying to find the optimal parameters that minimize the total costs of the system. However, the difficulty of this approach is the computation of the penalty cost, since it includes not only the direct cost of losing demand but also an indirect cost of losing customers goodwill and image of the company. For this reason, the most common approach in practical environments is to find the optimal parameters of the inventory system that guarantee the achievement of a target service level. In addition, this approach is more attractive to practitioners as it is easier for them to interpret the fill rate they strive to offer their customers [26]. However, to implement the service approach it is first required to have appropriate expression to compute accurately the service level because small errors in the calculation of the service level might have an important impact on the determination of optimal parameters [27]. In this sense, Table 4 presents the optimal value of s and S obtained with the classic and the proposed method when demand follows a Poisson distribution with $\lambda$ = 1 and L = 2. Note that the cells shaded in grey do not meet the hypothesis of only one outstanding order launched. As can be

Table 2. Average and standard deviation of errors between $\beta_{Classic}$ and $\beta_{Sim}$.

| | Error (%) | |
| --- | --- | --- |
| $\lambda$ | Average | Standard Deviation |
| 0.1 | 0.026 | 0.031 |
| 0.5 | 1.406 | 0.673 |
| 0.75 | 2.478 | 0.709 |
| 1 | 3.132 | 0.822 |
| 1.25 | 3.311 | 1.251 |
| 1.5 | 4.764 | 7.750 |

**Table 3. Base-stock level computed by $\beta_{Classic}$ and $\beta_{Exact}$ with Poisson demand with $\lambda = 1$, $s = 2$, $L = 2$.**

| S | $\beta_{Classic}$ | $\beta_{Exact}$ |
|---|---|---|
| 5 | 0.85 | 0.79 |
| 6 | 0.88 | 0.83 |
| 7 | 0.90 | 0.86 |
| 8 | 0.92 | 0.88 |
| 9 | 0.93 | 0.89 |
| 10 | 0.94 | 0.91 |
| 11 | 0.94 | 0.92 |
| 12 | 0.95 | 0.92 |
| 13 | 0.95 | 0.93 |
| 14 | 0.96 | 0.93 |
| 15 | 0.96 | 0.94 |
| 16 | 0.96 | 0.94 |
| 17 | 0.97 | 0.95 |

seen, there are different combination of $s$ and $S$ that achieve the same target fill rate. For example, a target fill rate of 0.95 can be achieved when $s = 2$ and $S = 13$ or when $s = 3$ and $S = 8$ according to the classic method. However, the real value of the fill rate with these parameters is 92.9% and 92.8%, respectively. Then, if these combinations are used as optimal parameters, the system might incur unexpected stockout situations, which lead to higher stockout costs of the system. In fact, the optimal parameters to achieve a 0.95 target fill rate are $s = 3$ and $S = 11$.

## Conclusions

This paper proposes, for the first time, an exact method for the calculation of the fill rate when the inventory is managed by the continuous review reorder point, base-stock $(s, S)$ policy in a context of lost sales and discrete demand. As opposed to the classic approach $(\beta_{Classic})$, the

**Table 4. Base-stock level and ROP computed by $\beta_{Classic}$ and $\beta_{Exact}$ with Poisson demand with $\lambda = 1$ and $L = 2$.**

| | Classic | | | | | | | | | | |
|---|---|---|---|---|---|---|---|---|---|---|---|
| s / S | 5 | 6 | 7 | 8 | 9 | 10 | 11 | 12 | 13 | 14 | 15 |
| 1 | 0.779 | 0.815 | 0.841 | 0.860 | 0.876 | 0.888 | 0.898 | 0.906 | 0.914 | 0.920 | 0.925 |
| 2 | 0.847 | 0.881 | 0.902 | 0.917 | 0.928 | 0.937 | 0.943 | 0.949 | **0.953** | 0.957 | 0.96 |
| 3 | | | 0.948 | **0.958** | 0.965 | 0.97 | 0.973 | 0.976 | 0.979 | 0.981 | 0.982 |
| 4 | | | | | 0.985 | 0.988 | 0.989 | 0.991 | 0.992 | 0.993 | 0.993 |
| 5 | | | | | | | 0.996 | 0.997 | 0.997 | 0.998 | 0.998 |
| 6 | | | | | | | | | 0.999 | 0.999 | 0.999 |
| 7 | | | | | | | | | | | 1 |
| | Exact | | | | | | | | | | |
| s / S | 5 | 6 | 7 | 8 | 9 | 10 | 11 | 12 | 13 | 14 | 15 |
| 1 | 0.733 | 0.772 | 0.801 | 0.823 | 0.841 | 0.855 | 0.867 | 0.878 | 0.886 | 0.894 | 0.901 |
| 2 | 0.794 | 0.835 | 0.861 | 0.881 | 0.895 | 0.906 | 0.915 | 0.923 | 0.929 | 0.934 | 0.939 |
| 3 | | | 0.912 | 0.928 | 0.939 | 0.946 | **0.952** | 0.957 | 0.961 | 0.964 | 0.967 |
| 4 | | | | | 0.968 | 0.973 | 0.977 | 0.979 | 0.981 | 0.983 | 0.985 |
| 5 | | | | | | | 0.99 | 0.991 | 0.992 | 0.993 | 0.994 |
| 6 | | | | | | | | | 0.997 | 0.997 | 0.998 |
| 7 | | | | | | | | | | | 0.999 |

exact method ($\beta_{Exact}$) takes explicit account of the presence of undershoots; consequently, it is not assumed that the replenishment order is launched when the inventory position exactly reaches the reorder point, but rather when it is equal to or lower than it.

Section 5 analyses the performance of $\beta_{Classic}$ and $\beta_{Exact}$ as opposed to a simulated fill rate. Results show that $\beta_{Classic}$ introduces a bias into the computation of the metric since it overestimates the fill rate. This performance was to be expected if we consider that neglecting the undershoots implies assuming that we have more on-hand stock available to meet future demand during the lead time than we actually do. Deviations presented by $\beta_{Classic}$ are more important when using it to compute the base-stock of the system, since small differences in the fill rate become higher when dealing with stock levels. The result of this article has a very important practical implication in regards to the design of (*s*, *S*) systems. In practice, the base-stock of the policy is determined to guarantee a target fill rate. If the method used to calculate the fill rate is one which systematically overestimates its real value, as occurs in the case of $\beta_{Classic}$, the established base-stock is lower than that required to reach the target service level and therefore the system will be less protected against possible stockouts than the inventory managers might expect.

As the Introduction section points out, neglecting the undershoots at the reorder point has important implications in inventory management. After analysing the results of this paper, it is clear that: (i) the impact of considering a replenishment order as constant and equal to *s-S* may not be ignored; (ii) it is necessary to take into account the variability of the replenishment cycle for accurate computation of the fill rate and the parameters of the (*s*, *S*) system; (iii) the mathematical complexity of the computation of the probability of every stock level when the reorder point is reached is acceptable for the benefit of an accurate fill rate estimation. The method presented in this paper eliminates the problems presented by the classical method, $\beta_{Classic}$, and thus entails a robust non-skewed method to exactly calculate the fill rate in the discrete demand and lost sales context, and therefore contributes to the operational research as the only exact method available in this context.

Limitations of this study mainly focus on the hypotheses regarding the i.i.d. demand condition. While this is a common assumption when dealing with theoretical inventory models, extending these methods such as the one addressed in this article to demand contexts that do not satisfy the i.i.d. condition may be beneficial for practical settings. Finally, the exact method derived in this paper, in addition to determining the customer service performance of the inventory management system, can be used as a constraint in deriving optimal stock policy values. This application of the exact method proposed in this article constitutes a challenging further research of this work.

## Supporting information

**S1 Table. $\beta_{Classic}$ $\beta_{Exact}$ and $\beta_{Sim}$ for the data from Table 1.**
(XLSX)

**S1 Appendix. Calculation of the transition matrixes $\overline{\overline{M_{\tau}}}$ and $\overline{\overline{M_L}}$.**
(DOCX)

**S2 Appendix. Derivation of the demand distribution from the beginning of the cycle until ROP.**
(DOCX)

## Acknowledgments

The authors would like to thank the anonymous reviewers for their work and invaluable contributions.

## Author Contributions

**Conceptualization:** Ester Guijarro, Eugenia Babiloni, Manuel Cardós.

**Formal analysis:** Ester Guijarro, Eugenia Babiloni, Manuel Cardós.

**Funding acquisition:** Ester Guijarro, Eugenia Babiloni.

**Investigation:** Ester Guijarro, Eugenia Babiloni.

**Methodology:** Ester Guijarro, Eugenia Babiloni, Manuel Cardós.

**Supervision:** Eugenia Babiloni.

**Writing – original draft:** Ester Guijarro, Eugenia Babiloni.

**Writing – review & editing:** Ester Guijarro, Eugenia Babiloni.

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
