## [Decision Letter · Decision Letter 0]

19 Oct 2021

PONE-D-21-14224On the estimation of the fill rate for the continuous ( s, S ) inventory system for the lost sales contextPLOS ONE

Dear Dr. Eugenia,

Thank you for submitting your manuscript to PLOS ONE. After careful consideration, we feel that it has merit but does not fully meet PLOS ONE’s publication criteria as it currently stands. Therefore, we invite you to submit a revised version of the manuscript that addresses the points raised during the review process.

I recommend that it should be revised taking into account the changes requested by the reviewers. Since the requested changes include valuable and constructive reviews, I would like to give you a chance to revise your manuscript. The revised manuscript will undergo the next round of review by two reviewers.

We look forward to receiving your revised manuscript.

Kind regards,

Baogui Xin, Ph.D.

Academic Editor

PLOS ONE

Journal Requirements:

When submitting your revision, we need you to address these additional requirements. 1. Please ensure that your manuscript meets PLOS ONE's style requirements, including those for file naming. The PLOS ONE style templates can be found at https://journals.plos.org/plosone/s/file?id=wjVg/PLOSOne_formatting_sample_main_body.pdf and https://journals.plos.org/plosone/s/file?id=ba62/PLOSOne_formatting_sample_title_authors_affiliations.pdf 2. We note that the grant information you provided in the ‘Funding Information’ and ‘Financial Disclosure’ sections do not match.  When you resubmit, please ensure that you provide the correct grant numbers for the awards you received for your study in the ‘Funding Information’ section. 3. In your Data Availability statement, you have not specified where the minimal data set underlying the results described in your manuscript can be found. PLOS defines a study's minimal data set as the underlying data used to reach the conclusions drawn in the manuscript and any additional data required to replicate the reported study findings in their entirety. All PLOS journals require that the minimal data set be made fully available. For more information about our data policy, please see http://journals.plos.org/plosone/s/data-availability. Upon re-submitting your revised manuscript, please upload your study’s minimal underlying data set as either Supporting Information files or to a stable, public repository and include the relevant URLs, DOIs, or accession numbers within your revised cover letter. For a list of acceptable repositories, please see http://journals.plos.org/plosone/s/data-availability#loc-recommended-repositories. Any potentially identifying patient information must be fully anonymized. Important: If there are ethical or legal restrictions to sharing your data publicly, please explain these restrictions in detail. Please see our guidelines for more information on what we consider unacceptable restrictions to publicly sharing data: http://journals.plos.org/plosone/s/data-availability#loc-unacceptable-data-access-restrictions. Note that it is not acceptable for the authors to be the sole named individuals responsible for ensuring data access. We will update your Data Availability statement to reflect the information you provide in your cover letter. 4. Please amend the manuscript submission data (via Edit Submission) to include author Eugenia Babiloni. 5. Please amend your authorship list in your manuscript file to include author 
María Eugenia 6. Please ensure that you refer to Figure 4 in your text as, if accepted, production will need this reference to link the reader to the figure. 7. We note you have included a table to which you do not refer in the text of your manuscript. Please ensure that you refer to Table 1 in your text; if accepted, production will need this reference to link the reader to the Table. 8. Please include a copy of Table 4 which you refer to in your text on page 13.

Reviewers' comments:

Reviewer's Responses to Questions

**Comments to the Author**

1. Is the manuscript technically sound, and do the data support the conclusions?

Reviewer #1: Partly

Reviewer #2: Partly

2. Has the statistical analysis been performed appropriately and rigorously? 

Reviewer #1: I Don't Know

Reviewer #2: N/A

3. Have the authors made all data underlying the findings in their manuscript fully available?

Reviewer #1: Yes

Reviewer #2: No

4. Is the manuscript presented in an intelligible fashion and written in standard English?

Reviewer #1: Yes

Reviewer #2: Yes

5. Review Comments to the Author

Reviewer #1: Referee report

In this draft, the authors have reported their efforts in proposing an exact method to estimate the fill rate in a base-stock (s,S) policy. Overall, this paper is good and it needs substantial changes. Following major points are recommended to revise the present article:

1- Literature survey should be improved by reviewing recently related studies. The references are too old.

2- You calculate the exact value of the fill rate by considering the value of S as a parameter. Can you set S as a decision variable and calculate its exact and optimal value considering the constant fill rate?

3- You may design and solve an inventory problem based on the base-stock policy, and use the proposed exact method to obtain the optimal value of s and S. In this way, the effectiveness of the exact method in reducing inventory system costs compared to the classical method can be seen.

4- In the section of “practical implications”, you refer to Table 4, but you provide the information in Table 3 (See line 283 and 287). This is also the case with Table 2 (See line 271).

5- In the conclusion section, you may explore the future extension of this proposed method mentioning major limitations of it.

Reviewer #2: Given possible undershoots at the reorder point, the authors proposed a method to accurately estimate the fill rate for the continuous (s, S) inventory system for the lost sales context. However, the paper has the following problems.

1. The undershoots at the reorder point might be only possible for lumpy and batch production situations. Now many companies use real-time systems and automate reorder process with built-in trigger functions, the undershoot issues might not be relevant. Thus, the authors need to clearly identify the limitations of the research.

2. For the exact approach, the authors assume that a Markov Chain transition process is followed. How about a non-Markov chain process？

6. PLOS authors have the option to publish the peer review history of their article (what does this mean?). If published, this will include your full peer review and any attached files.

Reviewer #1: No

Reviewer #2: No

---

## [Author Response · Author response to Decision Letter 0]

14 Nov 2021

RESPONSE TO ACADEMIC EDITOR AND REVIEWERS

The authors would firstly like to express our gratitude to the academic editor and the reviewers for all the comments they have made. All the changes made to the manuscript following their comments have contributed to an increase in the quality of our work. 

In this document, we answer all the comments made by them in the same order and showing where the changes have been made in the final version of the manuscript. Please note that our answers to the different concerns are written in blue and highlighted also in blue in the final version of the manuscript (Revised Manuscript with Track Changes file).

RESPONSE TO THE ACADEMIC EDITOR

Format has been updated to fulfill PLOS ONE’s style requirements. 

Furthermore in the new version of the manuscript the Appendix has been converted to S1 Appendix and S2 Appendix following the recommendations about Supporting Information; funding information has been removes from the acknowledgments; and format of headings, normal text, references, Equations, Figures and Tables has been reviewed.

The information has been duly updated.

According to the data policy, a new file (S1 Table) has been added to the submission to support the results of our paper. Also appendixes have been converted as Support information files.

4. Please amend the manuscript submission data (via Edit Submission) to include author Eugenia Babiloni.

The confusion in the names of the authors has been duly corrected, as Eugenia Babiloni and Maria Eugenia are the same person. Her name now appears correctly. Thank you for your suggestion.

5. Please amend your authorship list in your manuscript file to include author María Eugenia

See previous comment.

6. Please ensure that you refer to Figure 4 in your text as, if accepted, production will need this reference to link the reader to the figure.

Thank you for your comment. In the new version of the manuscript, Fig. 4 is quoted in the text.

7. We note you have included a table to which you do not refer in the text of your manuscript. Please ensure that you refer to Table 1 in your text; if accepted, production will need this reference to link the reader to the Table.

Thank you for your comment. In the new version of the manuscript Table 1 is quoted in the text.

8. Please include a copy of Table 4 which you refer to in your text on page 13.

Thank you for your comment. There was a typographical error in the numbering of the tables, which has been conveniently corrected. In the new version of the manuscript, Table 1, Table 2 and Table 3 are referenced properly in the document. In the new version of the manuscript, Table 4 is referred to the new example included under the “Practical implications” section.

RESPONSE TO REVIEWER 1

1- Literature survey should be improved by reviewing recently related studies. The references are too old.

The literature has been carefully reviewed. Older references have been deleted or replaced by more recent ones where possible.

In summary, the references that have been removed are (numbering of the old version of the manuscript):

7. Silver EA. A modified formula for calculating customer service under continuous inventory review. AIIE Trans. 1970;2: 241–245.

10. Karlin S, Scarf H. Studies in the Mathematical Theory of Inventory and Production. Stanford University Press, Stanford, CA; 1958.

17. Vincent P. Exact fill rates for items with erratic demand patterns. INFOR Inf Syst Oper Res. 1985;23: 171–181.

24. Hadley G, Whitin T. Analysis of Inventory Systems. Englewood Cliffs, NJ: Prentice-Hall; 1963.

References that have been added in the new version of the manuscript are:

4. Luo P, Bai L, Zhang J, Gill R. A computational study on fill rate expressions for single-stage periodic review under normal demand and constant lead time. Oper Res Lett. 2014;42: 414–417. doi:10.1016/j.orl.2014.07.004

9. Gutierrez M, Rivera FA. Undershoot and order quantity probability distributions in periodic review, reorder point, order-up-to-level inventory systems with continuous demand. Appl Math Model. 2021;91: 791–814.

26. Gutgutia A, Jha JK. A closed-form solution for the distribution free continuous review integrated inventory model. Oper Res. 2018;18: 159–186. doi:https://doi.org/10.1007/s12351-016-0258-5.

27. Babiloni E, Guijarro E. Fill rate: from its definition to its calculation for the continuous (s, Q) inventory system with discrete demands and lost sales. Cent Eur J Operarion Res. 2020;28: 35–43. doi:https://doi.org/10.1007/s10100-018-0546-7.

2- You calculate the exact value of the fill rate by considering the value of S as a parameter. Can you set S as a decision variable and calculate its exact and optimal value considering the constant fill rate?

The new version of the manuscript includes an explanation under “Practical implication” section (see Table 4) to illustrate that, given a target fill rate, we can find different combinations of optimal s and S. As this example shows, if the classic fill rate is used to determine the optimal values of the inventory policy (s and S), the systems is not actually meeting the target fill rate and, as a consequence, the systems is less protected against stockout situations than expected. (See the Response to Reviewers file for detais).

3- You may design and solve an inventory problem based on the base-stock policy, and use the proposed exact method to obtain the optimal value of s and S. In this way, the effectiveness of the exact method in reducing inventory system costs compared to the classical method can be seen.

Thank you for this comment. Following the Reviewer’s suggestion, we have included a new example in section “Practical implications” to illustrate how the proposed method can be used to find the optimal values of s and S when a target fill rate is given. As can be observed in the new Table 4, the use of the classic method may lead to unexpected stockout situations, and consequently to higher stock-out costs of the system. Please see comment 2 of this revision for explanation of the new version of the practical implications section.

In any case, we agree with the reviewer that it would be a very interesting contribution to derive a method for the optimal derivation of policy parameters, for which the objective of this paper is a major contribution. Following his recommendations, the application of the exact method proposed in this paper for the determination of control parameters of the inventory policy has been included as a further research as follows:

“[…].Finally, the exact method derived in this paper, in addition to determining the customer service performance of the inventory management system, can be used as a constraint in deriving optimal stock policy values. This application of the exact method proposed in this article constitutes a challenging further research of this work.”

4- In the section of “practical implications”, you refer to Table 4, but you provide the information in Table 3 (See line 283 and 287). This is also the case with Table 2 (See line 271).

Thank you for your comment. There was a typographical error in the numbering of the tables, which has been conveniently corrected. In the new version of the manuscript, Table 1,Table 2 and Table 3 are referenced properly in the document. In the new version of the manuscript, Table 4 is referred to the new example included under the “Practical implications” section.

5- In the conclusion section, you may explore the future extension of this proposed method mentioning major limitations of it.

Thank you for the suggestion. In the latest version of the manuscript, a paragraph on the limitations of this study has been introduced as follows:

“Limitations of this study mainly focus on the hypotheses regarding the i.i.d. demand condition. While this is a common assumption when dealing with theoretical inventory models, extending these methods such as the one addressed in this article to demand contexts that do not satisfy the i.i.d. condition may be beneficial for practical settings. […]”

RESPONSE TO REVIEWER 2

1. The undershoots at the reorder point might be only possible for lumpy and batch production situations. Now many companies use real-time systems and automate reorder process with built-in trigger functions, the undershoot issues might not be relevant. Thus, the authors need to clearly identify the limitations of the research.

Thank you for this comment. We have realized that the previous version of the manuscript did not clearly state the importance of the presence of undershoots in the (s, S) inventory policy since they appear when demand is not unit sized. Therefore, the problem of neglecting undershoots affects not only lumpy or batch production demand, but also non-intermittent demand. In fact, the illustrative examples used in this paper consider a demand rate of 1 unit and, even in that case, the classic method underestimates the real value of the fill rate due to the fact of neglecting undershoots. 

In order to avoid possible misunderstandings, we have better clarified the importance of undershoots in the “Introduction” section:

“[…] This policy entails an additional difficulty: to exactly reach the reorder point, the demand must be necessarily unitary. If this is not the case the inventory position may not be exactly at the reorder point, but a certain amount below it, called deficit or undershoot, which is difficult to quantify. Therefore common derivations of a (s, S) policy are based upon the assumption that the undershoots at the reorder point are negligible [6]. However, neglecting undershoots may lead to higher inventory costs or a lower service level than desired [8,9]. This is particularly true when managing intermittent, lumpy or slow-moving items for which the unitary demand assumption is not appropriate [10] although undershoots can also occur with non-intermittent demand as long as the unit sized demand assumption is not met.”

Also in “Problem description” section, at the end of the first paragraph:

“[…] However, in practical environments it is uncommon to find items whose demand is unitary, which makes it necessary to consider the presence of undershoots when determining optimal parameters of the inventory system.”

And at the end of “Results and discussion” section, when we comment the results of Table 2:

“[…] What can be seen in Table 2 is that when the demand rate increases, meaning it is more difficult to assume the unitary demand hypothesis, the average and deviation of the errors made by �Classic increase. However, this example illustrates that undershoots appears also for non-intermittent demand, which justifies that undershoots should be taken into account explicitly in the calculation of the fill rate.”

Furthermore, we have better explained the novelty of the proposed method, which is based on calculating the probability vector of the on-hand stock at ROP, instead of approximating the undershoots. In this way, our approach is valid both when undershoots appear and when they do not. In the new version of the manuscript, the third paragraph of “Introduction” section reads:

“[…] Our approach, on contrary, instead of deriving an approach of the distribution function of the undershoots, considers that the inventory position when the reorder point is reached may take any feasible value (i.e. from 0 to s). This entails not neglecting the undershoots at the reorder point and implies the difficulty of accurately calculating expected values of the on-hand stock at order delivery and when the reorder point is reached (exactly or exceeded). This approach makes our research innovative.”

Lastly, regarding the limitations and further research of this work, we have introduced a new paragraph at the end of the “Conclusions” section as follows:

“Limitations of this study mainly focus on the hypotheses regarding the i.i.d. demand condition. While this is a common assumption when dealing with theoretical inventory models, extending these methods such as the one addressed in this article to demand contexts that do not satisfy the i.i.d. condition may be beneficial for practical settings. Finally, the exact method derived in this paper, in addition to determining the customer service performance of the inventory management system, can be used as a constraint in deriving optimal stock policy values. This application of the exact method proposed in this article constitutes a challenging further research of this work.”

2. For the exact approach, the authors assume that a Markov Chain transition process is followed. How about a non-Markov chain process？

Thank you for pointing out that in the previous wording it looked like it might be a non-Markov process. We have partially rewritten the paragraph prior to equation (4) as follows:

“The stock level of policy (s, S) satisfies the Markov property, i.e. the future state only depends on the current situation. As a result, on-hand stock levels can be represented by a Markov process and the probability transition matrix of the on-hand stock between two consecutive replenishment cycles needs to be determined so that:”

---

## [Decision Letter · Decision Letter 1]

25 Jan 2022

On the estimation of the fill rate for the continuous ( s, S ) inventory system for the lost sales context

PONE-D-21-14224R1

Dear Dr. Babiloni,

We’re pleased to inform you that your manuscript has been judged scientifically suitable for publication and will be formally accepted for publication once it meets all outstanding technical requirements.

Kind regards,

Baogui Xin, Ph.D.

Academic Editor

PLOS ONE

Additional Editor Comments (optional):

Reviewers' comments:

Reviewer's Responses to Questions

**Comments to the Author**

1. If the authors have adequately addressed your comments raised in a previous round of review and you feel that this manuscript is now acceptable for publication, you may indicate that here to bypass the “Comments to the Author” section, enter your conflict of interest statement in the “Confidential to Editor” section, and submit your "Accept" recommendation.

Reviewer #2: All comments have been addressed

2. Is the manuscript technically sound, and do the data support the conclusions?

Reviewer #2: Yes

3. Has the statistical analysis been performed appropriately and rigorously? 

Reviewer #2: N/A

4. Have the authors made all data underlying the findings in their manuscript fully available?

Reviewer #2: Yes

5. Is the manuscript presented in an intelligible fashion and written in standard English?

Reviewer #2: Yes

6. Review Comments to the Author

Reviewer #2: The authors have addressed my comments in the previous round. I have no further comments in this round.

7. PLOS authors have the option to publish the peer review history of their article (what does this mean?). If published, this will include your full peer review and any attached files.

Reviewer #2: No

---

## [Editor Report · Acceptance letter]

31 Jan 2022

PONE-D-21-14224R1 

On the estimation of the fill rate for the continuous (*s, S*) inventory system for the lost sales context 

Dear Dr. Babiloni:

I'm pleased to inform you that your manuscript has been deemed suitable for publication in PLOS ONE. Congratulations! Your manuscript is now with our production department. 

Kind regards, 

on behalf of

Professor Baogui Xin 

Academic Editor

PLOS ONE